# Uncontrolled hypertension among hypertensive patients in Sub-Saharan Africa: A systematic review and meta-analysis

Tigabu Munye Aytenew[1]*, Amare Kassaw[2], Amare Simegn[3], Gedefaye Nibret Mihretie[3], Sintayehu Asnakew[4], Yohannes Tesfahun Kassie[5], Solomon Demis[6], Demewoz Kefale[2], Shegaw Zeleke[1], Worku Necho Asferie[6]

1 Department of Nursing, College of Health Sciences, Debre Tabor University, Debre Tabor, Ethiopia, 2 Department of Pediatrics and Child Health Nursing, College of Health Sciences, Debre Tabor University, Debre Tabor, Ethiopia, 3 Department of Reproductive Health, College of Health Sciences, Debre Tabor University, Debre Tabor, Ethiopia, 4 Department of Psychiatry, College of Health Sciences, Debre Tabor University, Debre Tabor, Ethiopia, 5 Department of Emergency Medicine and Critical Care Nursing, College of Health Sciences, Debre Tabor University, Debre Tabor, Ethiopia, 6 Department of Maternity and Neonatal Nursing, College of Health Sciences, Debre Tabor University, Debre Tabor, Ethiopia

* tigabumunye21@gmail.com

**Data Availability Statement:** All relevant data are within the paper and Supporting Information files.

## Abstract

### Introduction

Hypertension is a major global public health problem. It currently affects more than 1.4 billion people worldwide, projected to increase to 1.6 billion by 2025. Despite numerous primary studies have been conducted to determine the prevalence of uncontrolled hypertension and identify its associated factors among hypertensive patients in Sub-Saharan Africa, these studies presented inconsistent findings. Therefore, this review aimed to determine the pooled prevalence of uncontrolled hypertension and identify its associated factors.

### Methods

We have searched PubMed, Google Scholar, and Web of Science databases extensively for all relevant studies. A manual search of the reference lists of included studies was performed. A weighted inverse-variance random-effects model was used to compute the overall pooled prevalence of uncontrolled hypertension and the effect size of its associated factors. Variations across the included studies were checked using forest plot, funnel plot, $I^2$ statistics, and Egger's test.

### Results

A total of twenty-six primary studies with a sample size of 11,600 participants were included in the final meta-analysis. The pooled prevalence of uncontrolled hypertension was 50.29% (95% CI: 41.88, 58.69; $I^2$ = 98.98%; P<0.001). Age of the patient [AOR = 1.57: 95% CI: 1.004, 2.44], duration of diagnosis [AOR = 2.57: 95% CI: 1.18, 5.57], non-adherence to physical activity [AOR = 2.13: 95% CI: 1.15, 3.95], khat chewing [AOR = 3.83: 95% CI: 1.59, 9.24] and habitual coffee consumption [AOR = 10.79: 95% CI: 1.84, 63.24] were significantly associated with uncontrolled hypertension among hypertensive patients.

**Funding:** The authors received no specific funding for this work.

**Competing interests:** The authors have declared that no competing interests exist.

**Abbreviations:** CKD, Chronic kidney diseases; CVDs, Cardio-vascular diseases; DBP, Diastolic blood pressure; DM, Diabetes miletus; HIV, Human immune deficiency virus; JBI, Joanna Briggs Institute; LMICs, Low and middle-income countries; PRISMA, Preferred Reporting Items for Systematic Review and Meta-Analysis; SBP, Systolic blood pressure; SSA, Sub-Saharan Africa.

## Conclusions

The pooled prevalence of uncontrolled hypertension was considerably high. Older age, duration of diagnosis, non-adherence to physical activity, khat chewing and habitual coffee consumption were independent predictors of uncontrolled hypertension. Therefore, health professionals and other responsible stakeholders should encourage hypertensive patients to adhere to regular physical activity, and abstain from khat chewing and habitual coffee consumption. Early identification of hypertension and management of comorbidities is crucial, and it should be emphasized to control hypertension easily.

## Introduction

Hypertension is a major global public health problem [1–3]. It currently affects more than 1.4 billion people worldwide, and this is expected to project to 1.6 billion by 2025 [4,5]. Of these, 82% of the population live in low and middle-income countries(LMICs) [6]. The global prevalence of hypertension was estimated to be almost 40% among adults aged 25 years and above, and Africa accounts for the highest prevalence (46%) [7].

Nowadays, hypertension is increasingly emerging in LMICs where the health resources are scarce and stretched by a high burden of infectious diseases such as human immune deficiency virus (HIV), malaria, and tuberculosis, and where awareness and treatment levels of hypertension control are still very low [8]. The burden is high in Sub-Saharan Africa (SSA [2], affects nearly 25% of the adult population [9], and an estimated of 74.7 million individuals are hypertensive, and this number will also be projected to increase by 68% to 125.5 million individuals by 2025 [10–12].

Hypertension is the leading cause of cardiovascular diseases (CVDs) morbidity, mortality, and disabilities accounting for around 7.5 million deaths annually worldwide [4,5,9,13–16]. It doubles the risk of developing CVDs including coronary heart disease, congestive heart failure, stroke, renal failure, and peripheral arterial disease [1,10,12]. It is also an overwhelming global challenge and a third-ranked cause of disability-adjusted life year [17–19]. It is an essential contributor to the rising burden of CVDs in SSA which is expected to nearly double by 2030 [2].

Although hypertension is a preventable and modifiable risk factor of CVDs, its prevention and control have not yet received due attention in many developing countries [14,20,21]. If hypertension is left uncontrolled, it causes stroke, heart failure, dementia, coronary heart disease, peripheral vascular disease, renal impairment, retinal hemorrhage, and blindness, imposing severe financial and service burdens on the healthcare systems [18,22–25]. Though uncontrolled hypertension is a significant public health challenge in developed and developing countries [9,22], it has become higher in SSA than in Western countries over the past few decades, accounting for 70% of the total disease burden in the region [6].

Despite effective therapeutic options, hypertension remains uncontrolled in both developed and developing countries [6,12,26]. Although controlling hypertension is crucial in reducing hypertension-associated CVDs, it remains inadequately controlled in clinical practice [4,27]. Adequate control of hypertension requires the identification of factors associated with it [28]. Therefore, timely diagnosis, patient awareness, and access to effective treatment are important components in achieving hypertension control in the population [29].

Despite numerous primary studies have been made to determine the prevalence of uncontrolled hypertension and identify its associated factors among hypertensive patients in SSA, these studies presented inconsistent findings. Therefore, this systematic review and meta-analysis aimed to determine the overall pooled prevalence of uncontrolled hypertension and identify its associated factors.

## Methods

### Reporting and registration protocol

The Preferred Reporting Items for Systematic Reviews and Meta-Analyses (PRISMA) checklist [30] was used to report the results of this systematic review and meta-analysis (S1 Table in S1 File). The review protocol was registered with Prospero database (PROSPERO, 2023: CRD42023422846).

### Databases and search strategy

The adapted PICO format was used to retrieve the relevant primary studies. The adapted PICO consists of population (P), exposure (E), context (C), and outcome (O) as detailed below.

a. **Population**: Hypertensive patients in SSA

b. **Exposure**: Associated factors, risk factors, determinants, and predictors i.e. gender, increased age, lower educational level, duration of hypertension, comorbidity, duration of antihypertensive medications, durations of appointment, high cholesterol level, Khat chewing, habitual coffee consumption, salt intake, non-adherence to antihypertensive medications, non-adherence to physical activity, and obesity.

c. **Context (Setting)**: SSA, Ethiopia, Sudan, Kenya, Tanzania, Nigeria, Cameroon, and South Africa.

d. **Outcome:** Uncontrolled hypertension

Using the above adapted PICO, we developed the following review questions which were focused on retrieving all the relevant primary studies.

1. What is the prevalence of uncontrolled hypertension in SSA?

2. What are the factors associated with uncontrolled hypertension in SSA?

We have searched PubMed, Google Scholar, and Web of Science databases extensively for all available primary studies using the following search terms and phrases: (″Burden″ OR ″Magnitude″ OR ″Prevalence″ OR ″Incidence″) AND (″Uncontrolled hypertension″ OR ″Hypertension control″) AND (″Predictors″ OR ″Associated factors″ OR ″Risk factors″ OR ″Determinants″) AND ″Sub-Saharan Africa″. A manual search of the reference lists of included studies was performed. The searched studies were published between 2014 and 2023 in SSA and published in English language.

### Eligibility criteria

All observational (cross-sectional and retrospective cohort) studies that were conducted among adult (aged ≥18 years) hypertensive patients in SSA, and reported uncontrolled hypertension, and written in English language were included in the review. However, citations without abstracts and/or full texts, anonymous reports, editorials, systematic reviews and meta-analyses, outdated studies and qualitative studies were excluded from the review.

### Study selection

All the retrieved studies were exported to the EndNote version 7 reference manager to remove duplicate studies. Initially, two independent reviewers (TMA and AK) screened the titles and abstracts, followed by the full-text reviews to determine the eligibility of each study. The disagreement between the two reviews was solved through discussion.

### Data extraction

Two independent reviewers (TMA and WNA) extracted the data using a structured data extraction form. Whenever variations were observed in the extracted data, the phase was repeated. If discrepancies between the data extractors continued, the third reviewer (AK) was involved. The name of the first author, year of publication, country, study design, sample size, objective of the study, scale, statistical model, and effect size were collected.

### Primary outcome measure of interest

The primary outcome of interest of this review was the pooled prevalence of uncontrolled hypertension among hypertensive patients on anti-hypertensive medications in SSA.

### Operational definition of variables

Uncontrolled hypertension is defined as if SBP is $\geq$140mmHg and/or DBP $\geq$90mmHg for the general hypertensive population or if SBP is $\geq$130mmHg and/or DBP $\geq$80mmHg in patients with established diabetes miletus (DM) or chronic kidney disease (CKD) [14,24,25,28,31].

### Data analysis

The extracted data were exported to STATA version 17 for analysis. A weighted inverse-variance random-effects model [32] was used to compute the overall pooled prevalence of uncontrolled hypertension and the effect size of its predictors. The publication bias was checked by observing the symmetry of the funnel plot, and Egger's test with a p-value of <0.05 was also employed to determine a significant publication bias [33]. The percentage of total variation across studies due to heterogeneity was assessed using $I^2$ statistics [34]. The $I^2$ statistic of 0, 25, 50 and 75% values represented no, low, moderate, and high heterogeneity respectively [34]. A p-value of $I^2$ statistic <0.05 was used to declare a significant heterogeneity [35,36].

To identify the influence of a single study on the overall meta-analysis, sensitivity analysis was performed. Subgroup analysis based on the study area was employed to adjust the variations in the pooled estimate. A forest plot was used to estimate the effect of independent factors on the outcome variable, and a measure of association at 95% CI was also reported. The adjusted odds ratio (AOR) was the most frequently reported measure of association in the eligible primary studies, and a random-effects model was used to estimate the pooled AOR effect. The qualities of the studies were evaluated using JBI criteria. The findings were presented using figures, tables, and texts.

## Results

### Search results

A total of 1572 studies were retrieved from PubMed (n = 857), Google Scholar (n = 695), Web of Science (n = 05), a manual search (n = 12), and 03 studies from a research repository online library. After removing the duplicated studies (n = 64) and irrelevant studies based on their titles and abstracts (n = 1152), 356 studies were selected for full-text review. During full-text

review, 291 studies with no accessible full texts were removed. Of the remaining 65 studies, 39 studies were excluded (full texts were not written in English, different study settings, and the outcomes were not well defined).

Finally, 26 studies were extracted to determine the pooled prevalence of uncontrolled hypertension and identify its associated factors in SSA. We traced the PRISMA flow chart [37] to show the selection process from initially identified records to finally included studies (Fig 1).

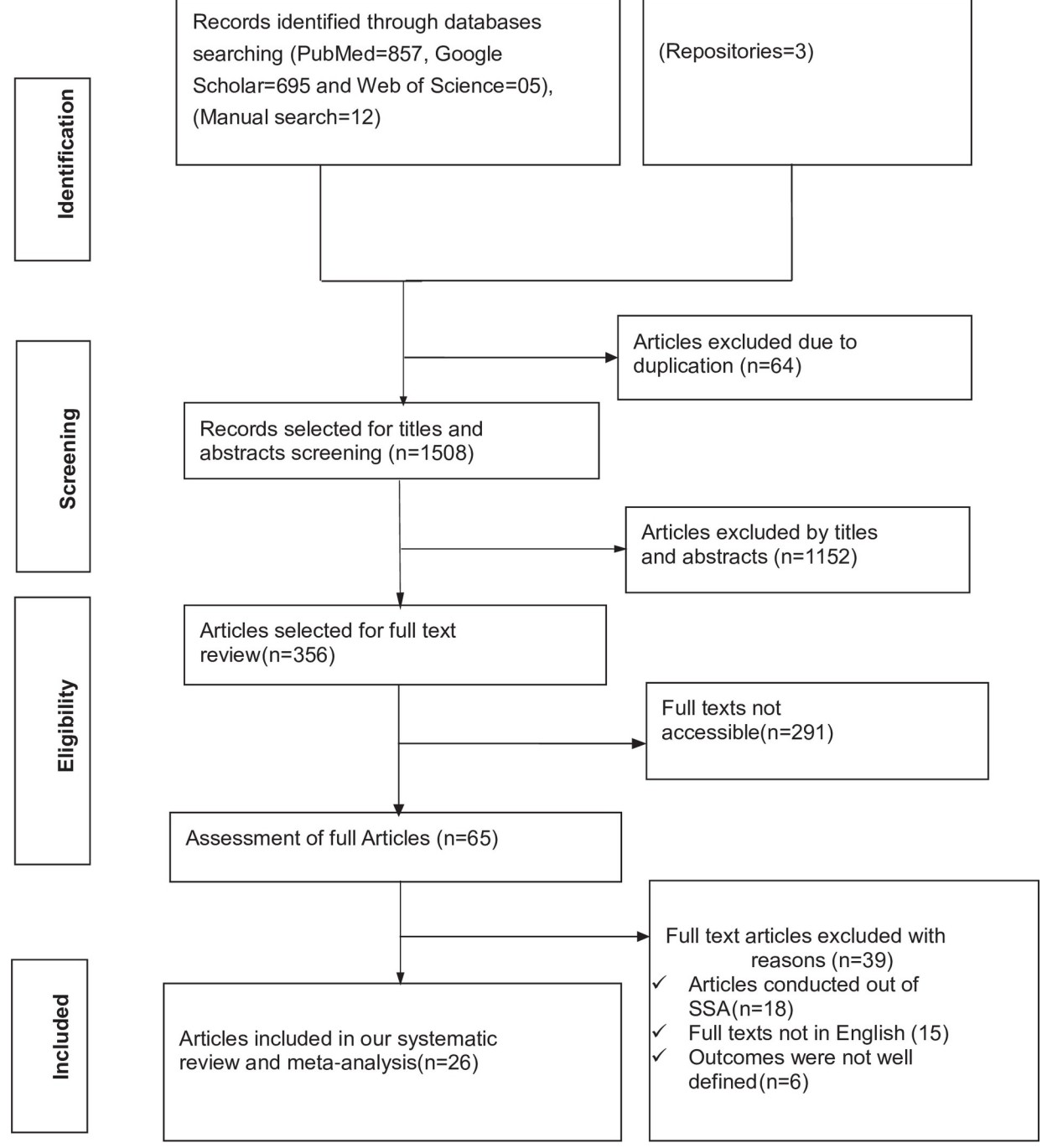

**Fig 1. PRISMA flow chart showing the studies selection process, 2023.**

**Table 1. General characteristics of the included primary studies, 2023.**

| ID | Author[Year] | Country | Study area | Study design | Sex | | Sample size | Response rate |
|----|--------------|---------|-----------|--------------|-----|-----|-------------|---------------|
| | | | | | M | F | | |
| | Abdisa L et al [2022] [6] | Ethiopia | HFSH, JGH, DCRH, SGH | CS | 220 | 195 | 415 | 98.60 |
| | Abdu O et al [2017] [1] | Ethiopia | UoGH | CS | 118 | 192 | 310 | 100 |
| | Abegaz TM et al [2017] [38] | Ethiopia | UoGH | CS | 227 | 316 | 543 | 93.90 |
| | Abegaz TM et al [2018] [28] | Ethiopia | UoGH | CS | 245 | 316 | 561 | 100 |
| | Aberhe W et al [2020] [9] | Ethiopia | Axsum CSH | CS | 177 | 219 | 396 | 100 |
| | Animut Y et al [2018] [10] | Ethiopia | UoGH | Cohort | 151 | 244 | 395 | 98.01 |
| | Antignac M et al [2018] [29] | SSA | SSA | CS | 874 | 1324 | 2198 | 100 |
| | Asgedom SW et al [2016] [39] | Ethiopia | JUSH | CS | 154 | 132 | 286 | 91.96 |
| | Berhe DF et al [2017] [44] | Ethiopia | AA & Tigray | Cohort | 346 | 551 | 897 | 93.0 |
| | Gebremikael GB etal[2019][22] | Ethiopia | Ayder CSH | CS | 156 | 164 | 320 | 100 |
| | Douglas KE et al [2018] [7] | Nigeria | Harcourt Hospital | CS | 194 | 223 | 417 | 98.58 |
| | Fekadu G et al [2020] [18] | Ethiopia | Nekemt RH | CS | 181 | 116 | 297 | 90.0 |
| | Lemessa F et al [2021] [14] | Ethiopia | Bale public hospitals | CS | 160 | 140 | 300 | 92.88 |
| | Lichisa GC et al [2014] [21] | Ethiopia | Adama Hospital | CS | 64 | 96 | 160 | 100 |
| | Magara GM et al [2022] [23] | Kenya | Thika level 5 hospital | CS | 134 | 260 | 394 | 100 |
| | Maginga J et al [2016] [11] | Tanzania | Bugando M/Centre | CS | 104 | 196 | 300 | 100 |
| | Masilela C et al [2020] [31] | S/Africa | Piet Retief hospital | CS | 61 | 268 | 329 | 100 |
| | Menanga A et al [2016] [12] | Cameron | Yaoundé hospitals | CS | 160 | 280 | 440 | 100 |
| | Muleta S et al [2017] [40] | Ethiopia | JU medical center | CS | 67 | 64 | 131 | 75.72 |
| | Negash AI et al [2023] [24] | Ethiopia | P/hospitals of Tigray | CS | 201 | 220 | 421 | 100 |
| | Omar SM et al [2018] [25] | Sudan | Gadarif hospital | CS | 146 | 234 | 380 | 100 |
| | Sheleme T et al [2022] [26] | Ethiopia | Bedele GH | CS | 132 | 87 | 219 | 100 |
| | Solomon M et al [2023] [41] | Ethiopia | Bishoftu H/facilities | CS | 249 | 149 | 398 | 100 |
| | Tesfaye B et al [2017] [42] | Ethiopia | JUSH | CS | 163 | 182 | 345 | 100 |
| | Teshome DF et al [2018] [43] | Ethiopia | DTGH | CS | 181 | 211 | 392 | 100 |
| | Yazie D et al [2018] [3] | Ethiopia | Zewditu M/Hospital | CS | 152 | 204 | 356 | 100 |

Abbreviations: AA, Addis Ababa; CS, cross-sectional; CSH, comprehensive specialized hospital; DCRH, Dill Chora referral hospital; HFSH, Hiwot Fana specialized hospital; JGH, Jugal general hospital; JU, Jimma University; JUSH, Jimma University specialized hospital; SGH, Sabian general hospital; UoGH, University of Gondar hospital and SSA, Sub-Saharan Africa.

## Characteristics of the included studies

The twenty-four studies [1,3,6,7,9,11,12,14,18,21–26,28,29,31,38–43] and two studies [10,44] were conducted using cross-sectional and retrospective cohort study designs respectively. Regarding geographical region, nineteen studies were found in Ethiopia [1,3,6,9,10,14,18,21,22,24,26,28,38–44], one study in Kenya [23], Nigeria [7], Cameroon [12], Sudan [25], South Africa [31], Tanzania [11], and selected SSA countries [29]. The total sample size of the included studies was 11,600, where the smallest sample size was 131 in Ethiopia (Jimma University Medical Center) and the largest sample size was 2,198 in selected SSA countries. The overall pooled prevalence of uncontrolled hypertension was obtained from all twenty-six included primary studies [1,3,6,7,9–12,14,18,21–26,28,29,31,38–44], while the data regarding the predictors of uncontrolled hypertension were obtained from the twenty-three studies [1,3,6,9–12,14,18,22,24–26,28,29,31,38–44] with a response rate ranging from 75.72 to 100% (Table 1).

## Quality assessment of the included studies

Two independent reviewers (TMA and WNA) appraised the quality of the included studies, and scored for the validity of the results. The quality of each study was evaluated using the Joanna Briggs Institute (JBI) quality appraisal criteria [45]. Twenty-four studies [1,3,6,7,9,11,12,14,18,21–26,28,29,31,38–43] and two studies [10,44] were assessed using JBI checklist for cross-sectional and cohort studies respectively. Thus, among the twenty-four cross-sectional studies, sixteen studies scored seven of the eight questions, 87.5% (low risk), five studies scored six of the eight questions, 75% (low risk), and the remaining three studies also scored five of the eight questions, 62.5% (low risk). Likewise, among the two cohort studies, one study scored eight of the ten questions, 80% (low risk), and the second cohort study scored seven of the ten questions, 70% (low risk) (S2 Table in S2 File).

Studies were considered low risk whenever they scored 50% and above on the quality assessment indicators. Therefore, from our quality appraisal, we generally found that all the included primary studies were reliable in their methodological quality scores, ranging from 5 to 7 of a total of 8 points for the cross-sectional studies, and 7 to 8 of a total of 10 points for the cohort studies. Thus, all the included studies [1,3,6,7,9–12,14,18,21–26,28,29,31,38–44] had high quality.

## Risk of bias assessment

The adopted assessment tool [46] was used to assess the risk of bias. It consists of ten items that assess four areas of bias: internal validity and external validity. Items 1–4 evaluate selection bias, non-response bias, and external validity. Items 5–10 assess measure bias, analysis-related bias, and internal validity. Accordingly, of the twenty-six included studies, twenty-two studies scored eight of ten questions, and the four studies scored seven of ten questions.

Studies were classified as ″low risk″ if eight and above of the ten questions received ″Yes″, as ″moderate risk″ if six to seven of the ten questions received ″Yes″ and as ″high risk″ if five or lower of the ten questions received ″Yes″. Therefore, all the included studies [1,3,6,7,9–12,14,18,21–26,28,29,31,38–44] had a low risk of bias (high quality) (S3 Table in S2 File).

## Meta-analysis

### Pooled prevalence of uncontrolled hypertension

Consequently, 26 eligible primary studies were included in the final meta-analysis. The prevalence of uncontrolled hypertension among hypertensive patients ranges from 11.42% in Ethiopia [28] to 77.40% in a study conducted among selected SSA countries [29], and the pooled prevalence of uncontrolled hypertension was 50.29% (95% CI:41.88, 58.69; $I^2$ = 98.98%; P<0.001) (Fig 2).

### Publication bias

The symmetry of the included primary studies on the funnel plot suggests the absence of a significant publication bias (Fig 3). The p-value of Egger's regression test (P = 0.458) also indicated the absence of publication bias.

### Investigation of heterogeneity

The percentage of $I^2$ statistics of the forest plot indicates a marked heterogeneity among the included studies ($I^2$ = 98.98%, P<0.001) (Fig 2). Hence, sensitivity and subgroup analyses were performed to minimize the heterogeneity.

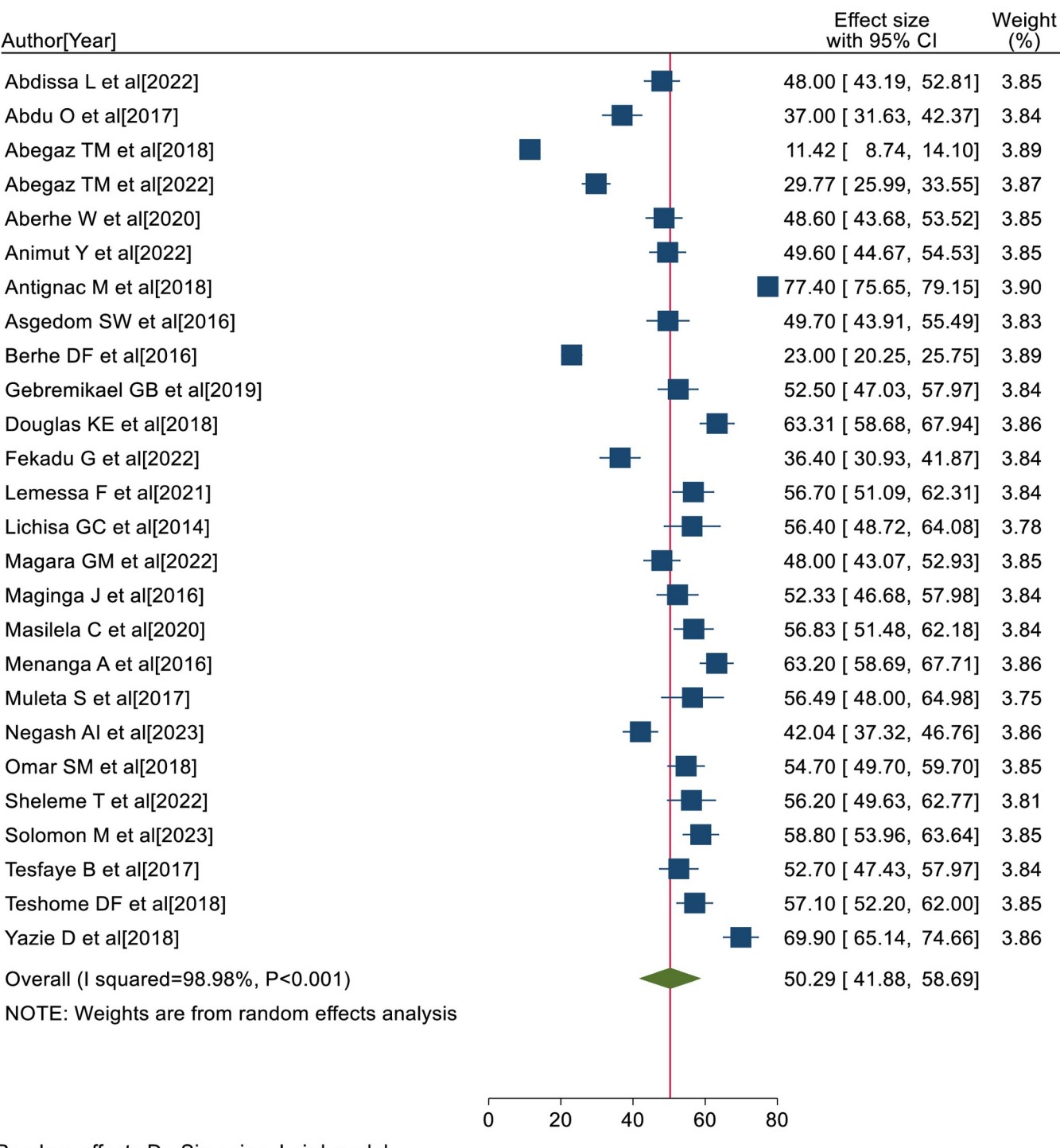

**Fig 2. Forest plot showing the pooled prevalence of uncontrolled hypertension in SSA, 2023.**

## Sensitivity analysis

To determine the influence of a particular primary study on the overall meta-analysis, we conducted a sensitivity analysis. The forest plot showed that the estimate of a single primary study

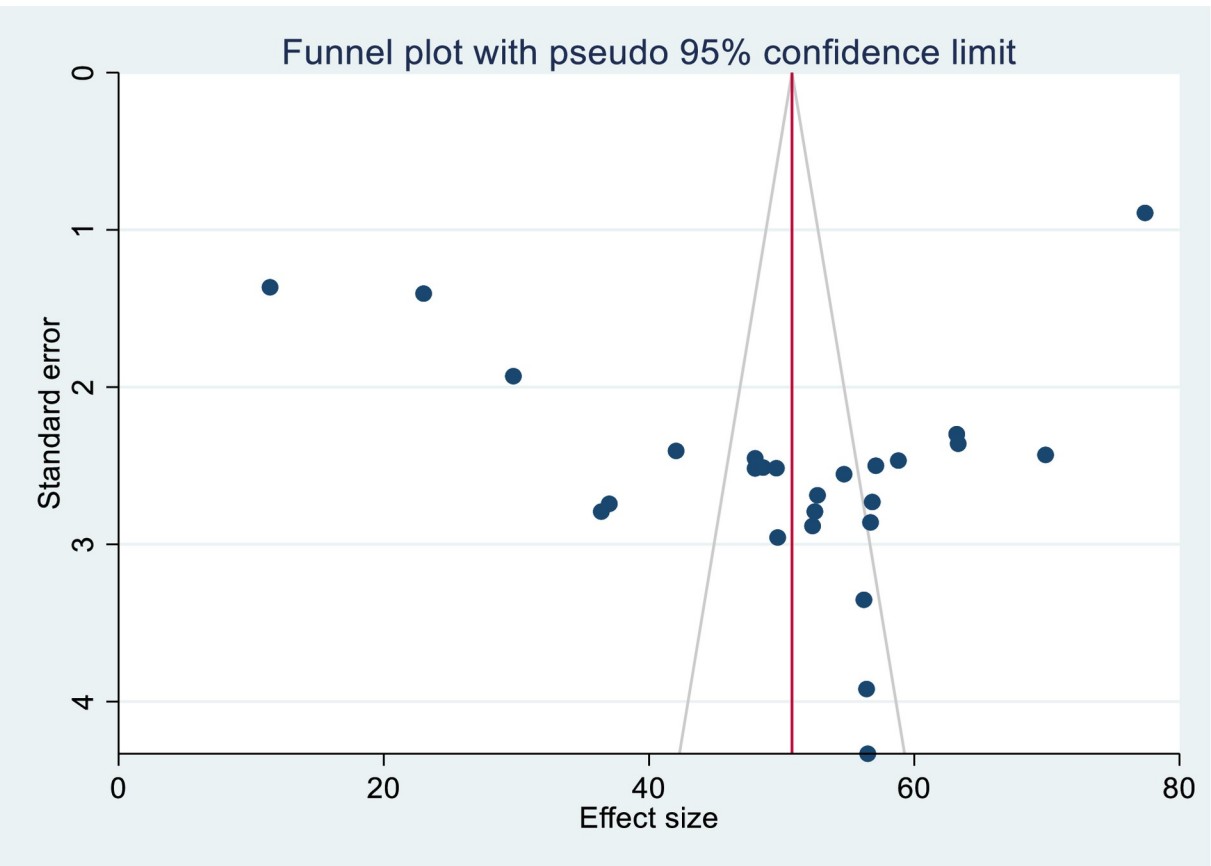

**Fig 3. Funnel plot showing the publication bias of uncontrolled hypertension in SSA, 2023.**

is closer to the combined estimate, which implied the absence of a single study effect on the overall pooled estimate. Thus, we declared that a single primary study has no significant impact on the overall outcome of the meta-analysis (Fig 4).

## Subgroup analysis using the study area

The highest pooled prevalence of uncontrolled hypertension was found among studies conducted out of Ethiopia [59.49: 95% CI: 49.47, 69.52; $I^2$ = 97.65%; P<0.001], followed by studies conducted in Ethiopia [46.88: 95% CI: 38.66, 55.09; $I^2$ = 98.29%; P<0.001].

## Factors associated with uncontrolled hypertension

The pooled analysis of the study finding showed that age of the patient [AOR = 1.57: 95% CI: 1.004, 2.44], duration of diagnosis [AOR = 2.57: 95% CI: 1.18, 5.57], non-adherence to physical activity [AOR = 2.13: 95% CI: 1.15, 3.95], khat chewing [AOR = 3.83: 95% CI: 1.59, 9.24] and habitual coffee consumption [AOR = 10.79: 95% CI: 1.84, 63.24] were significantly associated with uncontrolled hypertension.

Thus, patients who were habitual coffee consumers (>3 cups/day) were 10.79 times more likely to encounter uncontrolled hypertension compared to patients who were not habitual coffee consumers.

Patients who chew khat were also 3.83 times more likely to have uncontrolled hypertension than those who didn't [7,39,42].

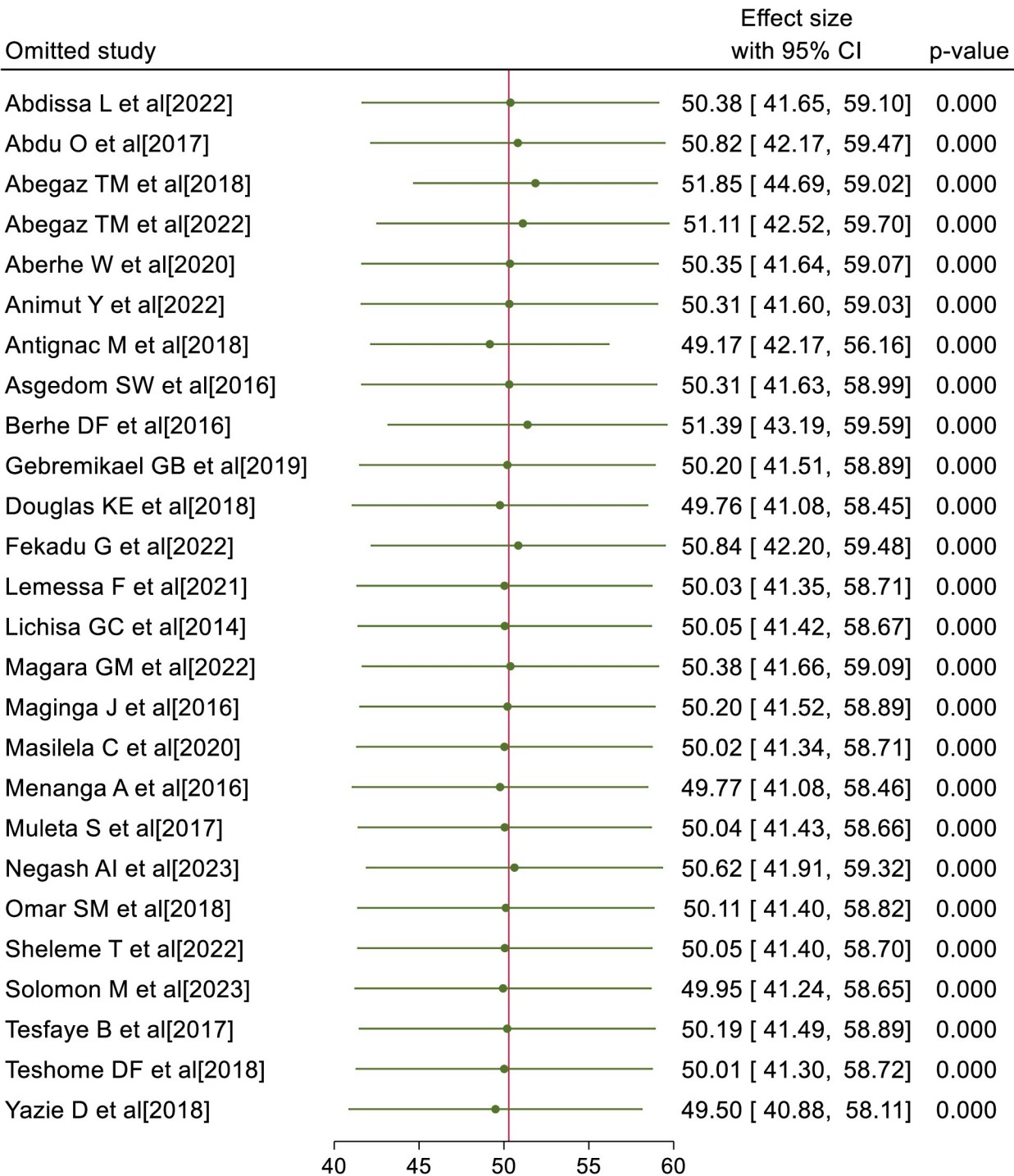

| Omitted study | Effect size with 95% CI | p-value |
|---|---|---|
| Abdissa L et al[2022] | 50.38 [ 41.65, 59.10] | 0.000 |
| Abdu O et al[2017] | 50.82 [ 42.17, 59.47] | 0.000 |
| Abegaz TM et al[2018] | 51.85 [ 44.69, 59.02] | 0.000 |
| Abegaz TM et al[2022] | 51.11 [ 42.52, 59.70] | 0.000 |
| Aberhe W et al[2020] | 50.35 [ 41.64, 59.07] | 0.000 |
| Animut Y et al[2022] | 50.31 [ 41.60, 59.03] | 0.000 |
| Antignac M et al[2018] | 49.17 [ 42.17, 56.16] | 0.000 |
| Asgedom SW et al[2016] | 50.31 [ 41.63, 58.99] | 0.000 |
| Berhe DF et al[2016] | 51.39 [ 43.19, 59.59] | 0.000 |
| Gebremikael GB et al[2019] | 50.20 [ 41.51, 58.89] | 0.000 |
| Douglas KE et al[2018] | 49.76 [ 41.08, 58.45] | 0.000 |
| Fekadu G et al[2022] | 50.84 [ 42.20, 59.48] | 0.000 |
| Lemessa F et al[2021] | 50.03 [ 41.35, 58.71] | 0.000 |
| Lichisa GC et al[2014] | 50.05 [ 41.42, 58.67] | 0.000 |
| Magara GM et al[2022] | 50.38 [ 41.66, 59.09] | 0.000 |
| Maginga J et al[2016] | 50.20 [ 41.52, 58.89] | 0.000 |
| Masilela C et al[2020] | 50.02 [ 41.34, 58.71] | 0.000 |
| Menanga A et al[2016] | 49.77 [ 41.08, 58.46] | 0.000 |
| Muleta S et al[2017] | 50.04 [ 41.43, 58.66] | 0.000 |
| Negash AI et al[2023] | 50.62 [ 41.91, 59.32] | 0.000 |
| Omar SM et al[2018] | 50.11 [ 41.40, 58.82] | 0.000 |
| Sheleme T et al[2022] | 50.05 [ 41.40, 58.70] | 0.000 |
| Solomon M et al[2023] | 49.95 [ 41.24, 58.65] | 0.000 |
| Tesfaye B et al[2017] | 50.19 [ 41.49, 58.89] | 0.000 |
| Teshome DF et al[2018] | 50.01 [ 41.30, 58.72] | 0.000 |
| Yazie D et al[2018] | 49.50 [ 40.88, 58.11] | 0.000 |

Random-effects DerSimonian–Laird model

**Fig 4. Sensitivity analysis of uncontrolled hypertension in SSA, 2023.**

Similarly, patients with five years and above of duration of diagnosis of hypertension were 2.57 times more likely to get uncontrolled hypertension than patients with less than five years of duration of diagnosis [18,40].

Moreover, patients who were non-adherent to physical activity were 2.13 times more likely to face uncontrolled hypertension than their counterparts [6,9,10,18,22,31,41,43].

Additionally, patients with the age of 50 years and above were around 1.57 times more likely to have the chance of getting uncontrolled hypertension than patients with the age of less than 50 years old [1,3,6,9,18,26,39,40,42–44].

## Discussion

In this review, the pooled prevalence of uncontrolled hypertension was 50.29% (95% CI: 41.88, 58.69); $I^2$ = 98.98%; P<0.001), which was higher than the finding of a systematic review and meta-analysis conducted in Ethiopia, 48% [4]. In addition, the mean prevalence of uncontrolled hypertension in this study was 49.55% [95% CI: 45.63, 53.47], which was lower than a study conducted in rural communities of South Asia, 58% [47]. This variation could be explained due to differences in study design, population characteristics, and sample size, and measurement methods. It's also possible that variations in healthcare systems and access to healthcare services across regions could play a significant role.

Likewise, patients who were habitual coffee consumers (>3 cups/day) were 10.79 times more likely to encounter uncontrolled hypertension compared to patients who were not habitual coffee consumers. This might be justified that caffeine has been hypothesized to raise blood pressure by several mechanisms, such as sympathetic stimulation, adenosine receptor antagonism, and elevated norepinephrine release by direct effects on the adrenal medulla, renal effects, and renin-angiotensin system activation; as a result, it may make the progress of treatment more challenging.

On the other hand, patients who chew khat were also 3.83 times more likely to have uncontrolled hypertension than patients who didn't chew. This is because Khat contains certain compounds that can affect the heart and blood vessels, causing them to function abnormally over time. As a result, individuals who regularly chew Khat may be at a greater risk of developing hypertension and other related health conditions.

Similarly, patients with five years and above duration of diagnosis of hypertension were 2.57 times more likely to get uncontrolled hypertension than patients with less than five years of duration of diagnosis. The gradual deterioration caused by the disease and a decrease in the patients' tendency to seek medical attention over time could be the reasons behind this situation. Essentially, the condition might be getting worse over time and the patients may not be seeking the necessary medical help as often as they should.

Moreover, patients who were non-adherent to physical activity were 2.13 times more likely to face uncontrolled hypertension than their counterparts. This could be justified because regular physical activity controls hypertension easily by enhancing heart and renal function and preventing weight gain.

Additionally, patients with the age of 50 years and above were also around 1.57 times more likely to have the chance of getting uncontrolled hypertension than patients with the age of less than 50 years old [1,3,6,9,18,26,39,40,42–44]. It could be explained that as age increases, it induces an increase in visceral fat and circulating leptin, which in turn increases the level of hypertension and makes it more challenging to control with treatment modalities.

## Strengths and limitations of the study

This review was the first study that combined the results of several studies conducted in Sub-Saharan Africa giving stronger evidence on uncontrolled hypertension. It was also able to include a large number of study participants (n = 11,600), which was much more than the sample sizes of the included primary studies. Though all of the studies are of good quality, most of

the included studies were cross-sectional, and only articles written in the English language were reviewed.

## Conclusions

The overall pooled prevalence of uncontrolled hypertension was considerably high. Moreover, the review showed that older age, duration of diagnosis, non-adherence to physical activity, khat chewing, and habitual coffee consumption were the independent predictors of uncontrolled hypertension. Therefore, health professionals and other responsible stakeholders should advance encouraging hypertensive patients to their weight management, increase their awareness/educational level, and take anti-hypertensive medications continuously as ordered. Early identification of hypertension and management of comorbidities among hypertensive patients is crucial, and it should be emphasized to control hypertension easily.

## Supporting information

**S1 File. S1 Table PRISMA checklist.**
(DOCX)

**S2 File. S2 and S3 Tables quality and risk of bias assessment of the included studies.**
(DOCX)

## Author Contributions

**Conceptualization:** Tigabu Munye Aytenew, Gedefaye Nibret Mihretie, Yohannes Tesfahun Kassie, Solomon Demis, Shegaw Zeleke.

**Data curation:** Tigabu Munye Aytenew, Amare Kassaw, Solomon Demis, Worku Necho Asferie.

**Formal analysis:** Tigabu Munye Aytenew, Amare Kassaw, Gedefaye Nibret Mihretie, Yohannes Tesfahun Kassie.

**Investigation:** Tigabu Munye Aytenew, Solomon Demis, Shegaw Zeleke, Worku Necho Asferie.

**Methodology:** Tigabu Munye Aytenew, Amare Kassaw, Gedefaye Nibret Mihretie, Sintayehu Asnakew, Yohannes Tesfahun Kassie, Shegaw Zeleke.

**Resources:** Shegaw Zeleke, Worku Necho Asferie.

**Software:** Amare Simegn.

**Validation:** Tigabu Munye Aytenew, Amare Simegn, Sintayehu Asnakew.

**Visualization:** Demewoz Kefale.

**Writing – original draft:** Tigabu Munye Aytenew, Amare Simegn.

**Writing – review & editing:** Tigabu Munye Aytenew, Sintayehu Asnakew, Demewoz Kefale.

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
