## [Decision Letter · Decision Letter 0]

14 Jul 2023

PONE-D-23-15308Uncontrolled Hypertension and Its Predictors among Hypertensive Patients in Sub-Saharan Africa: Systematic Review and Meta-AnalysisPLOS ONE

Dear Dr. Aytenew,

Thank you for submitting your manuscript to PLOS ONE. After careful consideration, we feel that it has merit but does not fully meet PLOS ONE’s publication criteria as it currently stands. Therefore, we invite you to submit a revised version of the manuscript that addresses the points raised during the review process. Please note that we have only been able to secure a single reviewer to assess your manuscript. We are issuing a decision on your manuscript at this point to prevent further delays in the evaluation of your manuscript. Please be aware that the editor who handles your revised manuscript might find it necessary to invite additional reviewers to assess this work once the revised manuscript is submitted. However, we will aim to proceed on the basis of this single review if possible.  Please submit your revised manuscript by Aug 28 2023 11:59PM. If you will need more time than this to complete your revisions, please reply to this message or contact the journal office at plosone@plos.org. Please include the following items when submitting your revised manuscript:A rebuttal letter that responds to each point raised by the academic editor and reviewer(s). You should upload this letter as a separate file labeled 'Response to Reviewers'.A marked-up copy of your manuscript that highlights changes made to the original version. You should upload this as a separate file labeled 'Revised Manuscript with Track Changes'.An unmarked version of your revised paper without tracked changes. You should upload this as a separate file labeled 'Manuscript'.

We look forward to receiving your revised manuscript.

Kind regards,

Dario Ummarino, PhD

Staff Editor

PLOS ONE

- https://doi.org/10.1111/tmi.13684

- https://doi.org/10.1371/journal.pone.0220710

- https://doi.org/10.2147/IBPC.S245068

- 10.23937/2474-3690/1510057

In your revision ensure you cite all your sources (including your own works), and quote or rephrase any duplicated text outside the methods section. Further consideration is dependent on these concerns being addressed.

Reviewers' comments:

Reviewer's Responses to Questions

**Comments to the Author**

1. Is the manuscript technically sound, and do the data support the conclusions?

Reviewer #1: Yes

2. Has the statistical analysis been performed appropriately and rigorously? 

Reviewer #1: Yes

3. Have the authors made all data underlying the findings in their manuscript fully available?

Reviewer #1: Yes

4. Is the manuscript presented in an intelligible fashion and written in standard English?

Reviewer #1: Yes

5. Review Comments to the Author

Reviewer #1: Title: Uncontrolled Hypertension and Its Predictors among Hypertensive Patients in Sub-Saharan Africa: Systematic Review and Meta-Analysis.

In this systematic review and meta-analysis, Aytenew et al aimed to determine the pooled burden of uncontrolled hypertension and identify its predictors. This is a commendable effort. However, there are several limitations, and some were not identified in the manuscript.

Firstly, why restrict the search to studies published between 2014 and 2023 in SSA? The reason for this should be explained in the discussion, because several important studies were omitted. In a previous metaanalysis on a similar subject, Addo et al included 37 studies published between 1975 and 2006 (Addo J, Smeeth L, Leon DA. Hypertension in sub-saharan Africa: a systematic review. Hypertension. 2007 Dec;50(6):1012-8. doi: 10.1161/HYPERTENSIONAHA.107.093336. Epub 2007 Oct 22.), which I believe would have enriched the present metaanalysis.

Secondly and importantly, 19/24 studies in the present metaanalysis were from Ethiopia (i.e. two-thirds of the studied population). This would make the title of the manuscript inappropriate. Thus the investigators may consider expanding the metaanalysis to span across more years and include studies from other countries. Otherwise the title should be modified to show that most of the data is from Ethiopia, for example "Uncontrolled Hypertension and Its Predictors among Hypertensive Patients in Ethiopia and selected Sub-Saharan African countries: Systematic Review and Meta-Analysis".

6. PLOS authors have the option to publish the peer review history of their article (what does this mean?). If published, this will include your full peer review and any attached files.

Reviewer #1: No

---

## [Author Response · Author response to Decision Letter 0]

26 Jul 2023

Editor Comment #01: Please ensure that your manuscript meets PLOS ONE's style requirements, including those for file naming. The PLOS ONE style templates can be found at

Authors’ response: Recognizing your comment, we have looked at the PLOS ONE style templates using the given link and we have ensured that our manuscript meets PLOS ONE's style requirements, including those for file naming. The requested corrections have been included throughout the revised version manuscript.

Editor Comment #02: We noticed you have some minor occurrence of overlapping text with the following previous publication(s), which needs to be addressed:

- https://doi.org/10.1111/tmi.13684

- https://doi.org/10.1371/journal.pone.0220710

- https://doi.org/10.2147/IBPC.S245068

Authors’ response: We are convinced of this comment, and we have revised these overlapping texts in the manuscript accordingly.

Editor Comment #03: In your revision ensure you cite all your sources (including your own works), and quote or rephrase any duplicated text outside the methods section. 

Authors’ response: Accepting your valuable comment, we have revised the duplicated texts outside the method section of the manuscript.

Editor Comment #04: Please include a separate caption for each figure in your manuscript.

Authors’ response: Recognizing your comment, we included a separate caption for each figure in the manuscript.

Reviewer # 1:

Reviewer # 1 comment and suggestion #01: Title: Uncontrolled Hypertension and Its Predictors among Hypertensive Patients in Sub-Saharan Africa: Systematic Review and Meta-Analysis.

In this systematic review and meta-analysis, Aytenew et al aimed to determine the pooled burden of uncontrolled hypertension and identify its predictors. This is a commendable effort. However, there are several limitations, and some were not identified in the manuscript.

Firstly, why restrict the search to studies published between 2014 and 2023 in SSA? The reason for this should be explained in the discussion, because several important studies were omitted. In a previous meta-analysis on a similar subject, Addo et al included 37 studies published between 1975 and 2006 (Addo J, Smeeth L, Leon DA. Hypertension in Sub-Saharan Africa: a systematic review. Hypertension. 2007 Dec;50(6):1012-8. doi: 10.1161/HYPERTENSIONAHA.107.093336. Epub 2007 Oct 22.), which I believe would have enriched the present meta-analysis.

Authors’ response: Thank you for your critical view of our manuscript! Today, the diagnostic modalities, treatment protocols, burden of chronic illnesses including hypertension and the living styles of the community are changed through time. Therefore, the findings of the 37 studies published between 1975 and 2006 might not represent the current situations (they are outdated). We have searched extensively all available primary studies published after this review across the region, unfortunately, the searched studies laid between the year 2014 and 2023. So, when we said “the search was restricted to studies published between 2014 and 2023”, to mean that unfortunately, the searched studies laid between 2014 and 2023 in SSA. 

Reviewer #1 comment and suggestion #02: Secondly and importantly, 19/24 studies in the present meta-analysis were from Ethiopia (i.e., two-thirds of the studied population). This would make the title of the manuscript inappropriate. Thus, the investigators may consider expanding the meta-analysis to span across more years and include studies from other countries. Otherwise, the title should be modified to show that most of the data is from Ethiopia, for example "Uncontrolled Hypertension and Its Predictors among Hypertensive Patients in Ethiopia and selected Sub-Saharan African countries: Systematic Review and Meta-Analysis". 

Authors’ response: We have accepted the given comment fully. Instead of including studies conducted outside SSA (aimed to determine the pooled prevalence of uncontrolled hypertension in SSA), we have fully accepted your feedback to modify the title, and we have revised the title directly based on the given direction.

---

## [Decision Letter · Decision Letter 1]

8 Oct 2023

PONE-D-23-15308R1Uncontrolled hypertension and its predictors among hypertensive patients in Ethiopia and selected Sub-Saharan African countries: Systematic review and Meta-analysisPLOS ONE

Dear Dr. Aytenew,

Thank you for submitting your manuscript to PLOS ONE. After careful consideration, we feel that it has merit but does not fully meet PLOS ONE’s publication criteria as it currently stands. Therefore, we invite you to submit a revised version of the manuscript that addresses the points raised during the review process.

We look forward to receiving your revised manuscript.

Kind regards,

Muktar Beshir Ahmed, PhD

Academic Editor

PLOS ONE

Reviewers' comments:

Reviewer's Responses to Questions

**Comments to the Author**

1. If the authors have adequately addressed your comments raised in a previous round of review and you feel that this manuscript is now acceptable for publication, you may indicate that here to bypass the “Comments to the Author” section, enter your conflict of interest statement in the “Confidential to Editor” section, and submit your "Accept" recommendation.

Reviewer #1: (No Response)

Reviewer #2: (No Response)

Reviewer #3: (No Response)

2. Is the manuscript technically sound, and do the data support the conclusions?

Reviewer #1: Partly

Reviewer #2: Partly

Reviewer #3: Partly

3. Has the statistical analysis been performed appropriately and rigorously? 

Reviewer #1: N/A

Reviewer #2: Yes

Reviewer #3: No

4. Have the authors made all data underlying the findings in their manuscript fully available?

Reviewer #1: Yes

Reviewer #2: Yes

Reviewer #3: Yes

5. Is the manuscript presented in an intelligible fashion and written in standard English?

Reviewer #1: Yes

Reviewer #2: No

Reviewer #3: No

6. Review Comments to the Author

Reviewer #1: I still believe that if the authors can remove "Ethiopia" from their search terms, they will get several additional articles to include in the analysis. Including "Ethiopia" in the search terms will surely introduce bias in the result.

They can take a look at the article cited below to see that the use of appropriate search terms could reveal a more robust result:

Ataklte F, Erqou S, Kaptoge S, Taye B, Echouffo-Tcheugui JB, Kengne AP. Burden of undiagnosed hypertension in sub-saharan Africa: a systematic review and meta-analysis. Hypertension. 2015 Feb;65(2):291-8. doi: 10.1161/HYPERTENSIONAHA.114.04394. Epub 2014 Nov 10. PMID: 25385758.

Reviewer #2: A recently published systematic review (SR) covers the same subject matter and research inquiry. If the authors intend to examine the prevalence of uncontrolled hypertension in low- and middle-income countries, I might consider reviewing a manuscript. Alternatively, to prevent duplicating published work, I recommend the authors to conduct a risk/ aetiology SR concerning risk factors for uncontrolled hypertension in Ethiopia.

Click the link below to see the similar articles published recently:

10.1186/s12872-020-01414-3

Uncontrolled hypertension in Ethiopia: a systematic review and meta-analysis of institution-based observational studies | BMC Cardiovascular Disorders | Full Text (biomedcentral.com)

Reviewer #3: Table 1 better present population characteristics, for examples age, race, sex, and more.

Table 1 does not correctly display the studys’ orresponding references. : Abegaz TM et al[2022] cannot be found in the reference list. There is a typo in "Abdissa." Animut's reference lists the year as "2018," but Table 1 indicates "2022."

It is better to add a reference index number to each study.

Several studies were conducted at UoGH around same time. Are they derived from the same patient cohort?

Line 256, the selection of the random-effects model should be based on the research question and study design. It is not selected for assessing the influence of a single primary study on the overall meta-analysis.

Line 255 sensitivity analysis:

The funnel plot is not a form of sensitivity analysis.The paragraph is not clearly written. Figure 3 (not Figure 4) does not appear to be symmetric.

Subgroup analyses by sample sizes or by the year 2020 are irrelevant and arbitrary. It would be more appropriate to perform subgroup analyses based on regions (such as Ethiopia, SE Ethiopia, NE Ethiopia) or similar age groups and other relevant characteristics.

Line 276 and predictors:

This section is not clearly written. For cross-sectional studies, the relationship is more likely to represent an association rather than a prediction. It is unclear how the adjusted odds ratios were calculated. Were these calculated from pooled analysis? Did the studies adjust for the same covariates in logistic regression? If not, the pooled results may be influenced by bias.

7. PLOS authors have the option to publish the peer review history of their article (what does this mean?). If published, this will include your full peer review and any attached files.

Reviewer #1: No

Reviewer #2: No

Reviewer #3: No

---

## [Author Response · Author response to Decision Letter 1]

18 Nov 2023

Dear Editors and Reviewers:

We sincerely appreciate the valuable comments and suggestions you raised. The thorough review helped immensely in the shaping of the manuscript. The comments and suggestions have been closely followed and revisions have been made accordingly. The following are the questions extracted from the Editors and Reviewers’ comments along with our summarized responses. Thank you very much for your constructive comments. We tried to inculcate your comments and questions as described below. The changes will be attached with

Title: Uncontrolled hypertension among hypertensive patients in Sub-Saharan Africa: Systematic review and Meta-analysis.

Authors:

TM: tigabumunye21@gmail.com

AK: amarekassaw2009@gmail.com

AS: amaresimegn99@gmail.com

GN: gedefayen@gmail.com

SA: sintie579@gmail.com

YT: tesfahunyohannes08@gmail.com

SD: solomondemis@gmail.com

DK: demewozk@yahoo.com

SZ: shegawzn@gmail.com

WN: workunecho@gmail.com

Reviewer #1:

Reviewer #1 comment and suggestion #01: I still believe that if the authors can remove "Ethiopia" from their search terms, they will get several additional articles to include in the analysis. Including "Ethiopia" in the search terms will surely introduce bias in the result. They can take a look at the article cited below to see that the use of appropriate search terms could reveal a more robust result:

Ataklte F, Erqou S, Kaptoge S, Taye B, Echouffo-Tcheugui JB, Kengne AP. Burden of undiagnosed hypertension in Sub-Saharan Africa: A systematic review and meta-analysis. Hypertension. 2015 Feb; 65(2):291-8. Doi: 10.1161/HYPERTENSIONAHA.114.04394. Epub 2014 Nov 10. PMID: 25385758.

Authors’ response: Thank you for your critical view of our manuscript! We tried to search articles extensively using different MeSH terms, even by removing "Ethiopia" across Sub-Saharan African countries. During the search process, we have already searched the above cited article (Burden of undiagnosed hypertension in Sub-Saharan Africa: A systematic review and meta-analysis. Hypertension, 2015 Feb; 65(2):291-8), but it was not in line with our title (Uncontrolled hypertension…).

Reviewer #2:

Reviewer #2 comment and suggestion #01: A recently published systematic review (SR) covers the same subject matter and research inquiry. If the authors intend to examine the prevalence of uncontrolled hypertension in low- and middle-income countries, I might consider reviewing a manuscript. Alternatively, to prevent duplicating published work, I recommend the authors to conduct a risk/etiology SR concerning risk factors for uncontrolled hypertension in Ethiopia. Click the link below to see the similar articles published recently: 10.1186/s12872-020-01414-3

Uncontrolled hypertension in Ethiopia: A systematic review and meta-analysis of institution-based observational studies | BMC Cardiovascular Disorders | Full Text (biomedcentral.com).

Authors’ response: Definitely! Our primary objective was to estimate the pooled prevalence of uncontrolled hypertension among hypertensive patients in Sub-Saharan Africa, and we have also cited the above reference on reference number 14 in the manuscript. 

Reviewer #3:

Reviewer #3 comment and suggestion #01: Table 1 better present population characteristics, for examples age, race, sex, and more.

Authors’ response: Thank you for your valuable comment! We have revised the table based on the given direction, but the age of the target population was already operationalized in the manuscript (Age ≥18 years).

Reviewer #3 comment and suggestion #02: Table 1 does not correctly display the studys’ corresponding references. : Abegaz TM et al [2022] cannot be found in the reference list. There is a typo in "Abdissa." Animut's reference lists the year as "2018," but Table 1 indicates "2022." 

Authors’ response: Thank you for your critical view and constructive comment! We have revised the studys’ corresponding references in Table 1 accordingly.

Reviewer #3 comment and suggestion #03: It is better to add a reference index number to each study.

Authors’ response: Thank you for your valuable comment! We have put a reference index number to each study in the table.

Reviewer #3 comment and suggestion #04: Several studies were conducted at UoGH around same time. Are they derived from the same patient cohort?

Authors’ response: Thank you for your valuable comment! These studies were carried out at UoGH on different patient cohort within some time variation ranging from 2017 to 2022. 

Reviewer #3 comment and suggestion #05: Line 256, the selection of the random-effects model should be based on the research question and study design. It is not selected for assessing the influence of a single primary study on the overall meta-analysis.

Authors’ response: Accepting your valuable comment, we have revised it 

Reviewer #3 comment and suggestion #06: Line 255 sensitivity analysis:

The funnel plot is not a form of sensitivity analysis. The paragraph is not clearly written. Figure 3 (not Figure 4) does not appear to be symmetric.

Authors’ response: Thank you for your valuable comment! We have revised this paragraph. But the sensitivity analysis was located on fig 4 not fig 3, because fig 3 was already located above this section/paragraph stating publication bias.

Reviewer #3 comment and suggestion #07: Subgroup analyses by sample sizes or by the year 2020 are irrelevant and arbitrary. It would be more appropriate to perform subgroup analyses based on regions (such as Ethiopia, SE Ethiopia, NE Ethiopia) or similar age groups and other relevant characteristics.

Authors’ response: Accepting your valuable comment, we have done the subgroup analysis using the study setting/area.

Reviewer #3 comment and suggestion #08: Line 276 and predictors:

This section is not clearly written. For cross-sectional studies, the relationship is more likely to represent an association rather than a prediction. It is unclear how the adjusted odds ratios were calculated. Were these calculated from pooled analysis? Did the studies adjust for the same covariates in logistic regression? If not, the pooled results may be influenced by bias.

Authors’ response: Thank you for your constructive comments! Yes! For cross-sectional studies, the relationship is more of an association not a prediction. Therefore, we have revised this section based on the given direction. And the adjusted odds ratios were calculated from the pooled analysis. The studies were also adjusted for the same covariates in logistic regression.

---

## [Decision Letter · Decision Letter 2]

16 Jan 2024

PONE-D-23-15308R2Uncontrolled hypertension among hypertensive patients in Sub-Saharan Africa: A Systematic review and Meta-analysisPLOS ONE

Dear Dr. Aytenew,

Thank you for submitting your manuscript to PLOS ONE. After careful consideration, we feel that it has merit but does not fully meet PLOS ONE’s publication criteria as it currently stands. Therefore, we invite you to submit a revised version of the manuscript that addresses the points raised during the review process.

We look forward to receiving your revised manuscript.

Kind regards,

Muktar Beshir Ahmed, PhD

Academic Editor

PLOS ONE

Journal Requirements:

Additional Editor Comments:

We appreciate your effort in revising the search strategy. Please provide a detailed explanation of the search strategy in the revised manuscript. As mentioned in the response letter, it appears that you revised the search strategy using various MeSH terms and removing ‘Ethiopia’ to conduct an extensive search across Sub-Saharan African countries.

However, the results section indicates that the number of studies captured in the new search strategy for sub-Saharan Africa does not seem accurate. The new strategy did not capture additional studies, as evidenced in the results section where 19 out of 26 studies are from Ethiopia. This suggests that the key strategy may not have been updated correctly.  

Reviewers' comments:

Reviewer's Responses to Questions

**Comments to the Author**

1. If the authors have adequately addressed your comments raised in a previous round of review and you feel that this manuscript is now acceptable for publication, you may indicate that here to bypass the “Comments to the Author” section, enter your conflict of interest statement in the “Confidential to Editor” section, and submit your "Accept" recommendation.

Reviewer #1: All comments have been addressed

Reviewer #3: All comments have been addressed

2. Is the manuscript technically sound, and do the data support the conclusions?

Reviewer #1: (No Response)

Reviewer #3: (No Response)

3. Has the statistical analysis been performed appropriately and rigorously? 

Reviewer #1: (No Response)

Reviewer #3: (No Response)

4. Have the authors made all data underlying the findings in their manuscript fully available?

Reviewer #1: (No Response)

Reviewer #3: (No Response)

5. Is the manuscript presented in an intelligible fashion and written in standard English?

Reviewer #1: (No Response)

Reviewer #3: (No Response)

6. Review Comments to the Author

Reviewer #1: (No Response)

Reviewer #3: All of my comments were addressed.

The article is good for publication.

I have no more concerns.

7. PLOS authors have the option to publish the peer review history of their article (what does this mean?). If published, this will include your full peer review and any attached files.

Reviewer #1: No

Reviewer #3: No

---

## [Author Response · Author response to Decision Letter 2]

18 Jan 2024

Editor Comment #01: 

Authors’ response: Thank you for your critical view of our manuscript! We have reviewed and revised our reference list intensively. 

Editor Comment #02: 

Please provide a detailed explanation of the search strategy in the revised manuscript. As mentioned in the response letter, it appears that you revised the search strategy using various MeSH terms and removing ‘Ethiopia’ to conduct an extensive search across Sub-Saharan African countries.

However, the results section indicates that the number of studies captured in the new search strategy for sub-Saharan Africa does not seem accurate. The new strategy did not capture additional studies, as evidenced in the results section where 19 out of 26 studies are from Ethiopia. This suggests that the key strategy may not have been updated correctly. 

Authors’ response: Thank you for your valuable comment. We have provided a detailed explanation of the search strategy to get additional primary studies across Sub-Saharan African countries using various MeSH terms and removing ‘Ethiopia’ in the revised manuscript. However, we couldn’t get any more for this review. 

Editor Comment #03: 

While revising your submission, please upload your figure files to the Preflight Analysis and Conversion Engine (PACE) digital diagnostic tool, https://pacev2.apexcovantage.com/. PACE helps ensure that figures meet PLOS requirements. To use PACE, you must first register as a user. Registration is free. Then, login and navigate to the UPLOAD tab, where you will find detailed instructions on how to use the tool.

Authors’ response: Thank you! We have used the PACE tool to make our figures suitable for PLOS requirements.

---

## [Decision Letter · Decision Letter 3]

18 Mar 2024

Uncontrolled hypertension among hypertensive patients in Sub-Saharan Africa: A systematic review and meta-analysis

PONE-D-23-15308R3

Dear Dr. Aytenew,

We’re pleased to inform you that your manuscript has been judged scientifically suitable for publication and will be formally accepted for publication once it meets all outstanding technical requirements.

Kind regards,

Guoying Wang, MD, PhD

Academic Editor

PLOS ONE

Additional Editor Comments (optional):

Reviewers' comments:

Reviewer's Responses to Questions

**Comments to the Author**

1. If the authors have adequately addressed your comments raised in a previous round of review and you feel that this manuscript is now acceptable for publication, you may indicate that here to bypass the “Comments to the Author” section, enter your conflict of interest statement in the “Confidential to Editor” section, and submit your "Accept" recommendation.

Reviewer #1: All comments have been addressed

Reviewer #3: (No Response)

2. Is the manuscript technically sound, and do the data support the conclusions?

Reviewer #1: Yes

Reviewer #3: (No Response)

3. Has the statistical analysis been performed appropriately and rigorously? 

Reviewer #1: Yes

Reviewer #3: (No Response)

4. Have the authors made all data underlying the findings in their manuscript fully available?

Reviewer #1: Yes

Reviewer #3: (No Response)

5. Is the manuscript presented in an intelligible fashion and written in standard English?

Reviewer #1: Yes

Reviewer #3: (No Response)

6. Review Comments to the Author

Reviewer #1: (No Response)

Reviewer #3: All of my comments were addressed.

The article is good for publication.

I have no more concerns.

7. PLOS authors have the option to publish the peer review history of their article (what does this mean?). If published, this will include your full peer review and any attached files.

Reviewer #1: No

Reviewer #3: No

---

## [Editor Report · Acceptance letter]

9 May 2024

PONE-D-23-15308R3 

PLOS ONE

Dear Dr. Aytenew, 

I'm pleased to inform you that your manuscript has been deemed suitable for publication in PLOS ONE. Congratulations! Your manuscript is now being handed over to our production team.

Kind regards, 

on behalf of

Dr. Guoying Wang 

Academic Editor

PLOS ONE